# Prediction of photosynthesis in Scots pine ecosystems across Europe by a needle-level theory

Pertti Hari[1], Steffen Noe[2], Sigrid Dengel[3], Jan Elbers[4], Bert Gielen[5], Veli-Matti Kerminen[6], Bart Kruijt[4], Liisa Kulmala[1], Anders Lindroth[7], Ivan Mammarella[6], Tuukka Petäjä[6], Guy Schurgers[8], Anni Vanhatalo[1], Markku Kulmala[6], and Jaana Bäck[1]

[1]Institute for Atmospheric and Earth System Research INAR, Faculty of Agriculture and Forestry, Department of Forest Sciences, P.O. Box 27, FI-00014 University of Helsinki, Finland
[2]Estonian University of Life Sciences, Institute of Agricultural and Environmental Sciences, Department of Plant Physiology, Kreutzwaldi 1, EE-51014 Tartu, Estonia
[3]Lawrence Berkeley National Laboratory, Climate and Ecosystem Sciences Division, 1 Cyclotron Road 84-155, Mail Stop 074-0316, Berkeley, CA 94720-8118, USA
[4]Wageningen University and Research, P.O. Box 47, 6700AA Wageningen, Netherlands
[5]University of Antwerp, Department of Biology, 2610 Wilrijk, Belgium
[6]Institute for Atmospheric and Earth System Research INAR, Faculty of Science, P.O. Box 68, FI-00014, University of Helsinki, Finland
[7]Lund University, Department of Physical Geography and Ecosystem Sciences, 22362 Lund, Sweden
[8]University of Copenhagen, Department of Geosciences and Natural Resource Management, Øster Voldgade 10, 1350 Copenhagen, Denmark

*Correspondence to*: Pertti Hari (pertti.hari@helsinki.fi)

**Abstract.** Photosynthesis provides carbon for the synthesis of macromolecules to construct cells during growth. This is the basis for the key role of photosynthesis in the carbon dynamics of ecosystems and in the biogenic $CO_2$ assimilation. The development of eddy covariance measurements for ecosystem $CO_2$ fluxes started a new era in the field studies of photosynthesis. However, the interpretation of the very variable $CO_2$ fluxes in evergreen forests has been problematic especially in transition times such as the spring and autumn. We apply two theoretical needle-level equations that connect the variation in the light intensity, stomatal action and the annual metabolic cycle of photosynthesis. We then use these equations to predict the photosynthetic $CO_2$ flux in five Scots pine stands located from northern timberline to Central Europe. Our result has strong implications on our conceptual understanding of the effects of the global change on the processes in boreal forests, especially of the changes in the metabolic annual cycle of photosynthesis.

# 1 Introduction

A large number of eddy-covariance (EC) measuring stations have been constructed into forests, peat lands, grasslands and agricultural fields (e.g., Baldocchi et al 2000). These stations have provided valuable insights into carbon and energy balances of various ecosystems, but the net fluxes measured with EC do not yield detailed information about the actual processes determining these fluxes. Therefore, an important step forward would be to connect the measured energy and carbon fluxes with the processes taking place in the vegetation and soil. In this way, one would obtain improved understanding of the changes in the metabolism and structure of ecosystems generated by the present global change.

The modeling of EC fluxes has received strong attention. The statistical approaches connect measured fluxes with environmental factors typically using rather simple 'big-leaf' models where parameters are determined from ecosystem-scale EC data (Landsberg and Waring, 1997; Peltoniemi et al., 2015). More theory-driven modeling approaches are based on knowledge of plant metabolism, and account for the structure of the considered ecosystem. For instance, the widely used model by Farquhar et al. (1980) is based on sound physiological knowledge on biochemical reactions, and it has been coupled with description of stomatal conductance to account for the effects of partial closure of stomata on leaf-scale photosynthesis and transpiration rate (Cowan and Farquhar, 1977; Collatz et al., 1991; Leuning, 1995; Mäkelä et al., 2004; Katul et al., 2010; Medlyn et al., 2011; Dewar et al., 2018). These coupled photosynthesis-stomatal conductance models are now widely adopted in vegetation and climate modelling (Chen et al., 1999; Krinner et al., 2005; Sitch et al., 2008; Lin et al., 2015), and also commonly evaluated against measured EC fluxes (Wang et al., 2007). The upscaling from leaf to ecosystem scale is done either using 'big-leaf' approaches (dePury and Farquhar, 1997; Wang and Leuning, 1998), or by incorporating the impacts of vertical canopy structure on microclimatic drivers, solar radiation in particular, via multi-layer models of different complexity (Leuning, 1995; Baldocchi and Meyers, 1998).

The seasonal onset and decline of photosynthesis is closely following the temperature history, although in the short term and during the growing season photosynthesis follows primarily light (e.g. Kolari et al., 2007). Duursma et al. (2009) analysed the sensitivity in modeled stand photosynthesis (GPP) across six coniferous forests in Europe, using a photosynthesis model with submodels for light attenuation within the canopy and optimal stomatal control. They concluded that stand GPP was related to several aggregated weather variables, especially to the change in the effective temperature sum or mean annual temperature at the sites. They also concluded that quantum yield was the most influential parameter on annual GPP, followed by a parameter controlling the seasonality of photosynthesis and photosynthetic capacity. This is in line with our approach to include the light and temperature changes to the activity of the photosynthetic machinery in the model predicting stand-scale photosynthesis.

It is well known already for decades that photosynthesis converts atmospheric $CO_2$ to organic intermediates and finally to sucrose in green foliage, and involves both biochemical and physical processes. Biochemistry operates at sub-cellular scale by

the actions of several essential molecules: pigment-protein complexes that capture the energy from light and simultaneously split water molecules; thylakoid membrane pumps and electron carriers that produce ATP (adenosine triphosphate) and NADPH (nicotinamide adenine dinucleotide phosphate) with the captured energy, and finally enzymes in the Calvin cycle that produce organic acids (phosphoglyceric acid) from atmospheric $CO_2$ utilizing the ATP and NADPH (Calvin et al., 1950; Arnon

et al., 1954a; Arnon et al., 1954b; Mitchell, 1961; Farquhar et al., 1980). These pigments, membrane pumps and enzymes form the photosynthetic machinery required for the biochemistry. The physical part of photosynthesis involves the consumption of $CO_2$ in mesophyll chloroplasts, which generates $CO_2$ flow from atmosphere into chloroplasts via stomata by diffusion (Farquhar and von Caemmerer, 1982; Harley et al., 1992), and widens the scale of phenomena from molecular to the needle and shoot level. All C3 plants have a similar photosynthetic machinery that synthetizes sugars using light energy and

atmospheric $CO_2$. This common functional basis generates common regularities in the behaviour of photosynthesis. The aim of our paper is to study the role of these regularities in the behaviour of the photosynthetic $CO_2$ flux, observed in the measurements at one site, Värriö, and use the above concepts to analyse the EC flux data in several Scots pine stands across Europe (Fig. 1).

**2 Methods**

Our purpose in this paper is to show that in order to predict the annual dynamics in photosynthesis of evergreen conifers, both stomatal conductance and the physiological processes related to the inherent carbon assimilation and light absorbance, and - essentially - their synchronized functioning in the system are needed. Therefore, we involved both the biochemical and physical

processes into the question of seasonality in evergreen canopy photosynthesis. In order to do this in a robust way, we followed the Newton's approach in discovering a way to construct equations to describe the diurnal behaviour of photosynthesis utilising knowledge of light and carbon reactions in photosynthesis (Hari et al., 2014, 2017). First, we defined concepts and introduced the fundamental features of light and carbon reactions of photosynthesis, the action of stomata, and diffusion of $CO_2$ (axioms). We finalised the theoretical analysis with the conservation of mass and evolutionary argument that combine the dominating

features in the quantitative description of the system. In this way, we obtained an equation for the behaviour of photosynthesis of a leaf during a day ($p(I,E)_D$) that links the theoretical knowledge and climatic drivers (light, temperature, and $CO_2$ and water vapour concentration) to photosynthesis.

$$p(I,E)_D = \frac{(u_{opt}\, g_{max}\, C_a + r)\, b\, f(I)}{u_{opt}\, g_{max} + b\, f(I)} \qquad (1)$$

Here, $p$ is the rate of photosynthesis, $E$ is transpiration rate, $I$ is irradiation, $b$ is a parameter called the efficiency of photosynthesis, $g_{max}$ is a parameter introducing stomatal conductance when stomata are fully open, $r$ is the rate of respiration, and $u_{opt}$ is optimal degree of stomatal opening obtained from as solution of the optimisation problem of stomatal behaviour (Hari et al 2014, 2017). The photosynthetic light response curve is given as $f(I)$ (see e.g., Mäkelä et al., 2004). Parameter values and units are given in Table 1.

We then analysed the annual cycle of evergreen foliage photosynthesis, by using as an example the common Eurasian evergreen tree species, Scots pine (*Pinus sylvestris* L.), as an example. Importantly, there is a strong annual cycle in the concentrations of active pigments, membrane pumps and enzymes, generating the distinctive seasonality in photosynthesis of evergreen foliage (Pelkonen and Hari, 1980; Öquist and Huner, 2003; Ensminger et al., 2004). The changing state of the photosynthetic machinery over the course of a year is a characteristic feature determining the annual cycle of photosynthesis in coniferous trees, especially in mid and high latitudes experiencing seasonal temperature and irradiance changes. These state changes involve a regulation system that synthetizes and decomposes pigments, membrane pumps and enzymes in the photosynthetic machinery. We introduced the fundamental behaviour of synthesis and decomposition to clarify the relationship between synthesis and temperature, and linked the synthesis and decomposition with the state of the photosynthetic machinery, $S$. Our mathematical analysis resulted in a simple differential equation (Hari et al., 2017) that describes the behaviour of the state of this photosynthetic machinery:

$$\frac{dS}{dt} = Max\{0, a_1 (T + T_f)\} - a_2 S - a_3 Max\{(T_f - T) * I, 0\} \qquad (2)$$

Here, $T_f$ is the freezing temperature of needles, $T$ is the temperature, $S$ is the state of photosynthetic machinery and $a_1$-$a_3$ are parameters describing the annual cycle of photosynthesis. We combined the state of photosynthetic machinery with the equation describing the photosynthesis during a day (Eq. (1)) to obtain a description of the annual GPP dynamics $p(I, E)_A$ (Eq 3). Our theoretical thinking determines the structure of these two equations.

$$p(I, E)_A = \frac{(u_{opt}\, g_{max}\, C_a + r)\, a_4\, S\, f(I)}{u_{opt}\, g_{max} + a_4\, S\, f(I)} \qquad (3)$$

Here, $g_{max}$ is the stomatal conductance at times when stomata are open, $C_a$ is the $CO_2$ concentration in atmosphere, $u_{opt}$ is the seasonal modulated degree of optimal stomatal control and $a_4$ is a parameter.

We estimated the values of the parameters in Eqs. (1) and (2) by analysing shoot-scale measurements of the $CO_2$ exchange in evergreen Scots pine made during four years at our measuring station SMEAR I in Värriö, Northeastern Finland. To gain robust results, we used 130 000 measurements of photosynthetic $CO_2$ flux made with chambers. We found that Eqs. (1) and (2) together predicted photosynthesis very successfully, explaining about 95 % of the variance in the measured $CO_2$ flux at the shoot level (Hari et al., 2017).

The EC methodology provides the mean $CO_2$ flux during some time interval, usually 30 min. In the case of a forest stand, the measured flux combines the photosynthesis of trees and of other vegetation growing on the site and, in addition, the respiration of plants and soil microbes. We extracted the ecosystem $CO_2$ flux generated by photosynthesis by removing respiration from the measurements with standard methods (Reichstein et al., 2005). In this way, we obtain the ecosystem-scale GPP time-series for all sites. We describe the measuring sites in more details in the Supplement.

We explored the role of regularities described with Eqs (1-3) in explaining variation of observed GPP in European pine forests. Applying our equations dealing with the photosynthesis of one shoot to predict photosynthesis at ecosystem level omits numerous additional phenomena apparent on that scale. These include e.g. site-specific differences in the structure of shoots and canopy, adaption and acclimation of structure and metabolism to e.g. water availability, and extinction of light in the canopy, etc. These omitted phenomena generate noise in the prediction of photosynthesis at ecosystem level and consequently reduce goodness of fit of the prediction of GPP. Therefore, the transition from leaf to ecosystem level requires a rough description of the differences between shoot and ecosystem, and between ecosystems. We describe these differences with an ecosystem-specific scaling coefficient. As the first step of the prediction, we determined the values of the scaling coefficients from measurements done at each site during the year preceding the one we were aiming to predict. Thereafter we were able to predict the GPP in the five pine stands in Europe. We based our prediction utilising the two equations on the measured values of light, temperature and $CO_2$ and water vapour concentrations done in each site, on the parameter value obtained by the shoot-scale measurements in Värriö, and on the site-specific scaling coefficients determined from the eddy-covariance measurements done on the sites during the previous year. We developed a code in MatLab to perform the predictions.

## 3 Results

The predictions obtained for all measured Scots pine ecosystems were successful in describing the dynamic features of GPP (Fig. 2). The daily patterns of modeled photosynthetic $CO_2$ fluxes are very similar to the measured ones in each studied ecosystem throughout the photosynthetically active period. The predictions capture adequately the daily patterns: rapid

increase of GPP after sunrise, its saturation in the middle of the day, and its decline when the light intensity is decreasing towards evening. Clear proofs of its predictive power on a daily scale are the occasions when clouds reduce the light intensity to variable degrees, causing rapid variations in the $CO_2$ flux (Fig 2, Brasschaat day 186 and 187) and strong reduction in the $CO_2$ flux on days with heavy clouds (Fig 2, day 184 in Värriö and day 213 in Norunda).

The patterns found in the annual cycle of photosynthesis are very different at the different measurement sites in Europe. We defined the onset of photosynthesis at each site as the moment when the running mean of 14 days of photosynthetic $CO_2$ flux exceeds 20 % of the corresponding running mean in midsummer, and the moment of cessation of photosynthesis as the moment when the running mean of GPP has declined to 20 % of its summer time value. Our prediction of the timing of onset and

10 cessation of photosynthesis in the different measuring sites was quite successful, and the observed and predicted dates were very close to each other at all measurement sites (Fig. 3 panels A and B). Surprisingly, the parameter values in the differential equation dealing with the annual dynamics. i.e., the synthesis and decomposition of the photosynthetic machinery, obtained from shoot-scale measurements in Värriö, seemed to produce quite adequate predictions at ecosystem level in the other studied Scots pine stands although they are growing in very different climates.

The prediction power of GPP by our equations in five Scots pine ecosystems in Scandinavia and in Central Europe was higher than what we expected. The equations predicted successfully the rapid variations in all studied ecosystems, even though the residual variation was evidently a bit larger in the southern than in the northern ecosystems (Fig. 4). Our predictions using the parameters from Värriö explained about 80 % of the variance of photosynthetic $CO_2$ flux in the measured ecosystems. The

20 maximum proportion of explained variance was 93 % in SMEAR II and minimum 75 % in Brasschaat. Due to the quite large measuring noise of eddy-covariance measurements, about 10–30 % (Rannik et al., 2004; Richardson et al., 2006), it probably dominates the residuals, i.e. the difference between measured and predicted fluxes. We studied further the residuals as function of light, temperature, $CO_2$ and water vapour concentration (Fig. 4), but detected only minor systematic behaviour in the residuals, indicating that these factors were not determining the difference between the measured and predicted values. To

25 analyse the robustness of the results when scaled from leaf to stand scale, we also tested the difference between sites in the modelled and measured GPP when the ecosystem-specific scaling coefficient was based on the reported leaf area indexes, and these results (analysis not shown) indicate that the dynamics of ecosystem-level photosynthesis are rather independent of LAI values. This shows that the functional regularities determined in the model structure are able to capture the essential processes in the evergreen foliage photosynthesis.

## 4 Discussion and conclusions

Although the annual behaviour of carbon exchange in ecosystems is rather well documented as a phenomenon, we have found no theory/model that links the variations in environmental factors and the photosynthetic $CO_2$ flux of Scots pine ecosystems during a yearly cycle. Our results are in line with Duursma et al (2009) who tested the relative importance of climate, canopy structure and leaf physiology across a gradient of forest stands in Europe, and concluded that the annual dynamics of photosynthesis was closely connected to seasonal temperature variations and the temperature sums. However, their model explained only 62% of variation in annual GPP across site-years, due to their model structure which was more sensitive to soil moisture or leaf area changes.

Our result that the behaviour of measured gross primary production in Scots pine stands follows the same equations in a large area in Europe from the northern timber line to the strongly polluted areas in Central Europe near the southern edge of the Scots pine growing area opens new possibilities for investigating carbon budgets of evergreen forest ecosystems. The light and carbon reactions and the stomatal actions determine the daily behaviour of $CO_2$ flux between the Scots pine ecosystem and the atmosphere. Temperature has a dominating role in the dynamics of the annual cycle of photosynthesis.

The present global climate change stresses the importance to understand the ecosystem responses to increasing atmospheric $CO_2$ concentration and temperature. Equations 1 and 2 resulted in an adequate prediction of the GPP for all five studied Scots pine ecosystems. We can expect that the differential equation provides also adequate predictions of the photosynthetic response to a temperature increase in Lapland when the increase is smaller than the mean temperature difference between Värriö and Brasschaat, i.e. about 10 °C. Equations 1 and 2 provide also a prediction of the photosynthetic response of Scots pine ecosystems to increasing atmospheric $CO_2$ concentration, based on changes in carbon reactions of photosynthesis. The physiological basis of the photosynthetic response in the model is sound and, in addition, the residuals of our prediction show no clear trend as function of atmospheric $CO_2$ concentration (Fig. 4).

The prediction of daily and annual behaviour of photosynthesis based on the presented two equations was successful in five Scots pine ecosystems, expanding from northern timberline to Central Europe. The regularities observed in the shoot-scale measurements in Värriö seem to play a very important role in the photosynthetic $CO_2$ flux in evergreen Scots pine ecosystems across quite large geographical range. Our result provides some justification to think that there are also other common regularities in the behaviour of forests to be discovered.

**Data availability**

Data measured at the SMEAR I and II stations is available on the following website: http://avaa.tdata.fi/web/smart/. The data is licensed under a Creative Commons 4.0 Attribution (CC BY) license. Data measured at Norunda, Brasschaat and Loobos is available via ICOS Carbon Portal. Model codes can be obtained from Pertti Hari upon request (pertti.hari@helsinki.fi).

**Competing interests**

The authors declare that they have no conflict of interest.

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

Table 1. Symbols and parameters in model equations

| Name of parameter | Symbol | Unit | Notes |
|---|---|---|---|
| Rate of photosynthesis | p | $\mu$mol $CO_2$ m$^{-2}$ s$^{-1}$ | |
| Rate of transpiration | E | mmol $H_2O$ m$^{-2}$ s$^{-1}$ | |
| Photosynthetically active irradiation | I | $\mu$mol photons m$^{-2}$ s$^{-1}$ | |
| Efficiency of photosynthesis | b | $\mu$mol g m$^{-5}$ s$^{-1}$ | |
| Stomatal conductance when stomata are fully open | $g_{max}$ | mmol $H_2O$ m$^{-2}$ s$^{-1}$ | |
| Optimal degree of stomatal opening | $u_{opt}$ | unitless | |
| $CO_2$ concentration in ambient air | $C_a$ | g m$^{-3}$ | |
| Rate of respiration | r | $\mu$mol $CO_2$ m$^{-2}$ s$^{-1}$ | |
| Temperature | T | K | |
| State of photosynthetic machinery | S | unitless | |
| Parameters describing the annual cycle of photosynthesis, estimated using numeric methods (see Hari et al 2017) | a1 … a4 | | a1 = 10 <br> a2 = 0.065 <br> a3 = 2 <br> a4 = 1.15 * 10$^{-7}$ |

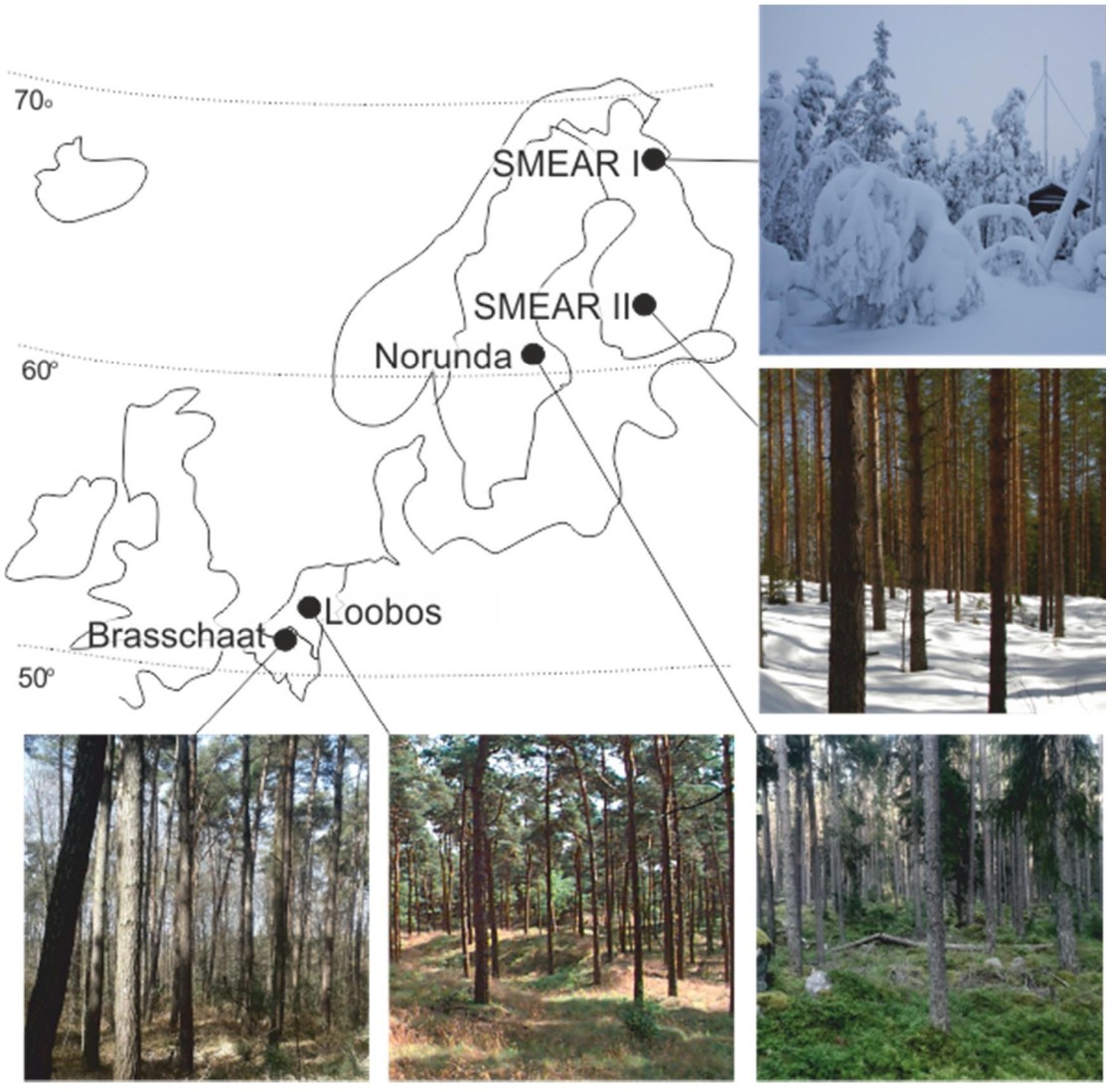

**Figure 1. The location of the measuring stations in Europe and photos of the stands. The photo of SMEAR I is taken around Christmas time, SMEAR II early spring, Norunda, Loobos and Brasschaat in summer time.**

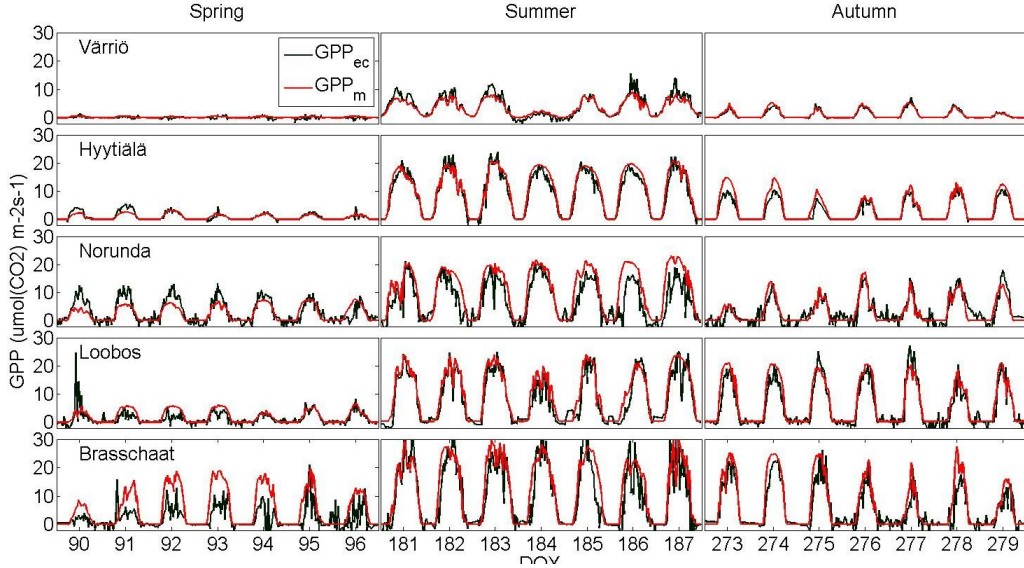

**Figure 2.** The measured (black) and predicted (purple)  photosynthetic $CO_2$ flux (GPP) between forest ecosystem and the atmosphere as function of time in five eddy-covariance measuring sites in Europe during a week in early spring, summer and autumn.

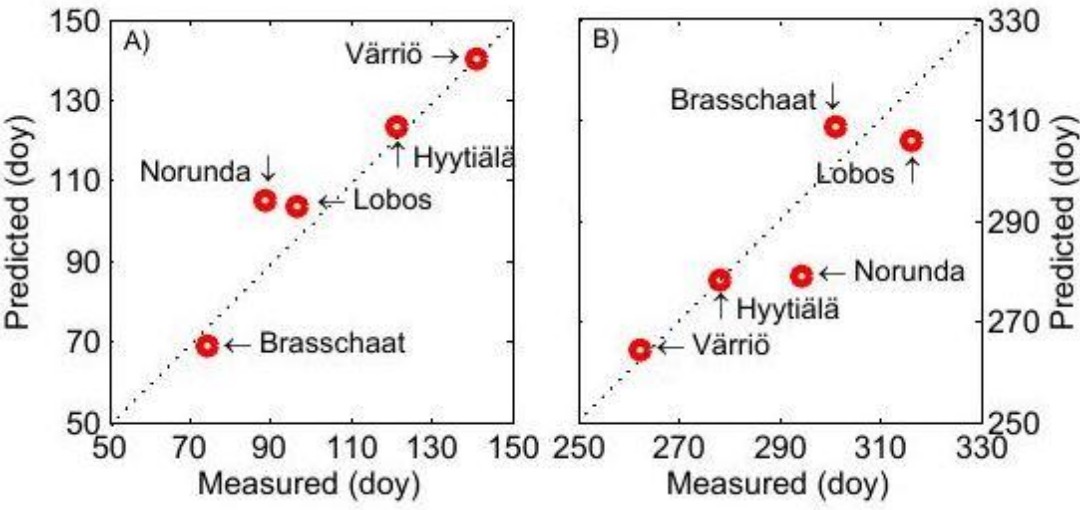

**Figure 3. A** The relationship between measured and predicted onset and cessation dates of photosynthesis in the five studied ecosystems, **B** the cessation dates of photosynthesis in the five ecosystems.

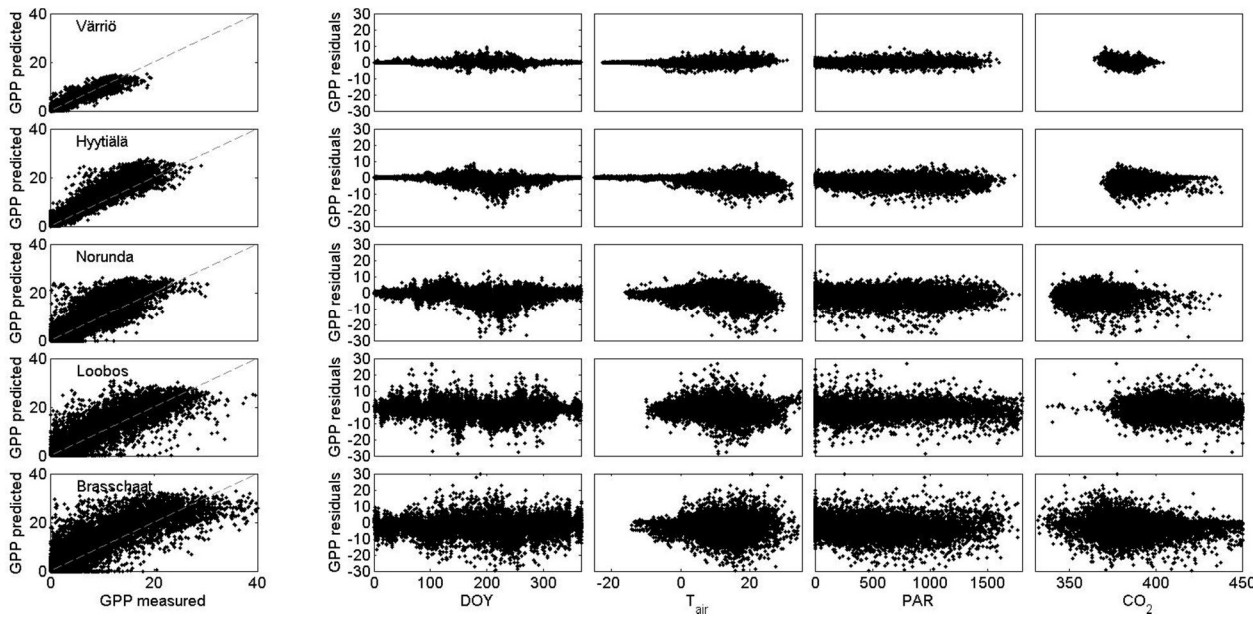

**Figure 4. The relationship between measured and predicted gross primary production (the first column). Columns 2-4 present the residuals as function of time, air temperature, photosynthetically active radiation and carbon dioxide**
10 **concentration.**

