# Peer review of "Prediction of photosynthesis in Scots pine ecosystems across Europe by a needle-level theory"

_Atmospheric Chemistry and Physics, 2017_

## Referee Comment (RC1) · Anonymous Referee #1 · 14 Sep 2017

The manuscript presents seasonal variation in half-hourly GPP estimates for five Scots pine stands from northern timberline to temperate central Europe. Eddy-covariance-based GPP estimates are compared with modeled fluxes. A leaf-level photosynthesis-stomatal conductance model, based on a theory of optimal stomatal behavior, is paired with a temperature- and light-driven 'state of photosynthetic machinery' model, which describes the seasonal changes in leaf physiology that drive those in their gas exchange. These (or similar) model structures have been tested in earlier publications. In the present study, the parameters of the leaf-level model are fitted to data collected from the northernmost stand. This single set of parameters is then used across the sites. The leaf-level flux is scaled up to stand using a site-specific scaling parameter that was derived using an independent dataset. The authors report on quite a remark-

able predictive power of the model across the five ecosystems ranging >10C in the mean annual temperature.

The findings of this manuscript are both very interesting and important and will most certainly trigger further research, but to be of high impact, the paper needs more work.

General comments:

1) The upscaling procedure deserves to be discussed in more detail. Because the leaf-level model is tested at the ecosystem scale, explaining the range of values of the 'ecosystem-specific scaling coefficient' is an essential part of assessing the role of 'common regularities in the behavior of photosynthesis' in ecosystem-atmosphere $CO_2$ exchange. In other words, when the modeled leaf-level flux is scaled to match measured ecosystem flux, does the scaling coefficient incorporate among-sites differences in canopy structure (leaf area and/or shoot structure) only? One could conjure a scenario in which, for example, both leaf area and photosynthetic efficiency change when moving from one stand to another. Why not compare estimates of GPP per unit leaf area across sites?

2) The structure of the paper would benefit from further streamlining. Related to the previous point, there is a range in the level of detail given, or depth of discussion, among various sections, which interferes with the flow of the paper. For example, the lack of consistent descriptions of the methods makes it difficult to follow (and to replicate) what was done. Also, it would be very helpful if all the parameters and drivers in all the equations were described and their units and fitted values (including the scaling coefficient) were given.

Specific comments:

P1L20-22: What do you mean by stable regularities? The study by Duursma et al. (2009, Tree Physiology 29, 621–639) appears relevant here.

P3L15: Do you mean conifers from high latitudes?

P4L22: What do you mean by 'differences in species' here?

P6L11-15: Please reduce repetition.

Fig.4: How do the residuals relate to soil moisture?

---

## Referee Comment (RC2) · Anonymous Referee #2 · 25 Sep 2017

The manuscript reported GPP estimated from eddy-covariance systems in five Scots pine vegetation in Europe. A leaf-level gas exchange model was used to explain the behavior of GPP in response to the environmental factors across the five study sites. The novelty in this study may be the consideration associated with the adjustment in photosynthetic machinery due to the changes in temperature in the proposed modeling framework. However, further elaboration on model development and the interpretation of model-data comparison is required.

General comments:

1. I am not able to follow the modeling framework. A separate section for the model description is necessary. The model derivation in detail and a list of variables and units should be also provided in the Supplement. Is the adjustment in photosynthetic

machinery due to the changes in temperature common for Scots pine? To my knowledge, the photosynthetic machinery in some species (e.g., Pinus edulis, Juniperus monosperma and Pinus taeda) even did not acclimate after long-term manipulation of precipitation and atmospheric CO2 concentration. The authors also pointed out that acclimation is omitted in the proposed model. However, how acclimation occurred at a longer time-scale is differentiated from the short-term changes in photosynthetic machinery needs further explanation.

2. To predict GPP across the five Scots pine stands from leaf-level model, a scaling coefficient was used to bridge the two largely separated spatial scales. The scaling coefficient for current year was estimated by data from previous year. This suggests that the scaling coefficient is dynamic (i.e., yearly). What would be the information from this yearly scaling coefficient? When the dynamic of photosynthetic machinery is only estimated from one site and subsequently used for the other four sites, how would you interpret the differences in the scaling coefficients across the five sites?

3. I am not sure if the proposed model can accommodate the effects of water-stressed condition in the soil on stomatal conductance especially when the authors mainly focus on the responses to light and CO2. In fact, how the differences in the environmental factors impact the behavior of GPP across the five sites is not discussed in the manuscript. If water-stressed condition in the soil is not explicitly considered in the leaf-level model, do we expect that this information is embedded in the scaling coefficient?

Specific comments:

1. P2L22 Definition of stable regularities is needed.

2. The order of Fig. 1 and 2 should be corrected to match the main text.

3. Comparison between measured and modeled S (i.e., the state of photosynthetic machinery) as well as related discussions should be provided.

4. P4L25 Description of up-scaling processes and the calculated scaling coefficient should be reported. Interpretation for the scaling coefficient is also required.

5. P5L13-16 Is it possible that the photosynthetic parameters for the five sites are actually different but this information is embedded in scaling coefficient?

6. P5L27-29 There are many models that can be used to predict stomatal conductance and then photosynthetic $CO_2$ flux in response to different environmental factors.

7. Discussion regarding different environmental conditions (e.g., temperature, precipitation, soil water status. . . . . . . . .) across the five sites should be included. To do so, time series of environmental factors for the five sites should be also provided when comparing the measured with predicted GPP (i.e., fig 1 or 2?).

---

## Author Comment (AC1) · 21 May 2018

**Final Author Comments for**

**Prediction of photosynthesis in Scots pine ecosystems across Europe by needle-level theory**

by Pertti Hari et al. in Atmos. Chem. Phys. Discuss.

We are grateful to the reviewers for their insightful and detailed comments and have below replied (in red) to each of them individually.

**Anonymous Referee #1**

**General comments:**

1) The upscaling procedure deserves to be discussed in more detail. Because the leaf-level model is tested at the ecosystem scale, explaining the range of values of the 'ecosystem-specific scaling coefficient' is an essential part of assessing the role of 'common regularities in the behavior of photosynthesis' in ecosystem-atmosphere  $CO_2$  exchange. In other words, when the modeled leaf-level flux is scaled to match measured ecosystem flux, does the scaling coefficient incorporate among-sites differences in canopy structure (leaf area and/or shoot structure) only? One could conjure a scenario in which, for example, both leaf area and photosynthetic efficiency change when moving from one stand to another. Why not compare estimates of GPP per unit leaf area across sites?

The most part of the "ecosystem –specific" behavior we describe in this paper is linked to temperature and introduced in the model via the S parameter. A minor part is dedicated to all other site-specific differences, LAI, moisture etc. The measure on a certain scale reflects the integrated or averaged value at that particular scale. Moving to larger scales is inevitably, accompanied by a loss in details. Moving from leaf level photosynthesis to EC data over a certain ecosystem's footprint leads to loosing of detailed information on the variation within single leaves/branches, and we gain a more integrated/averaged value of the underlying fundamental process.

Thus, our scaling coefficient incorporates all these differences between sites, including e.g., differences in fine structure of needles, the functional properties of the photosynthetic machinery, canopy structure often described with LAI, site fertility and other factors. The variation of LAI ultimately scales also with a temperature gradient and LAI and photosynthetic efficiency are linearly linked to each other. So dividing GPP by unit leaf area will only change the slope if we express the result in logarithmic units.

Our theory opens numerous interesting possibilities to study photosynthesis such as analyzing the GPP per unit leaf area. However, these questions are outside of the focus of this study, since we are aiming to describe the applicability of the fundamental principles of the annual dynamics of photosynthesis in a large eco-climatic scale, and we believe adding more details would not be very useful for this analysis.

2) The structure of the paper would benefit from further streamlining. Related to the previous point, there is a range in the level of detail given, or depth of discussion, among various sections, which interferes with the flow of the paper. For example, the lack of consistent descriptions of the methods makes it difficult to follow (and to replicate) what was done. Also, it would be very helpful if all the parameters and drivers in all the equations were described and their units and fitted values (including the scaling coefficient) were given.

Response: We have rewritten parts of the paper and structured it to improve the clarity. The details of the model and especially the fundamental concept of the annual cycle has been published in Hari, P., et al. (2017) Annual cycle of Scots pine's photosynthesis, Atmos. Chem. Phys., 17, 15045-15053,

https://doi.org/10.5194/acp-17-15045-2017), which is referred to in the text. A Table of parameters, units and fitted values was added (Table 1).

**Specific comments:**

P1L20-22: What do you mean by stable regularities? The study by Duursma et al. (2009, Tree Physiology 29, 621–639) appears relevant here.

Response: Thank you for pointing this out. With 'stable regularities' we mean here the fundamental and simple processes which can be scaled from smaller units to larger regions over the whole continent. Our approach is to develop here a model, which is robust enough to be applied for the annual variations of photosynthesis with very small number of environmental parameters and widely different climatic conditions.

We refer to the Duursma et al paper on p. 3 in the Introduction and on p. 6 in the discussion. Our results are in line with their analysis on conifer stand photosynthesis.

**P3L15: Do you mean conifers from high latitudes?**

Response: The annual cycle is a consequence of seasonal patterns in physical drivers of photosynthesis, most importantly temperature and irradiation. These changes cause a state change in the photosynthetic machinery from active to inactive state (e.g., transition from summer to winter) or vice versa. In some ecosystems the driver can be water availability which causes similar state changes. The text was revised to clarify that here we mean the trees in mid- and high latitudes experiencing seasonal temperature and irradiance changes.

P4L22: What do you mean by 'differences in species' here?

Response: This was a typo from previous version, we are grateful that the referee pointed it out. It was removed from the text.

P6L11-15: Please reduce repetition.

Response: Thanks for suggesting this, the text was rewritten and repetitions removed. We hope it now has a better structure and flow.

Fig.4: How do the residuals relate to soil moisture?

Response: The very small residuals in Figure 4 indicate that water stress or other environmental factors actually are of minor importance when the model incorporates the dynamic features of the annual dynamics of photosynthetic machinery.

**Anonymous Referee #2**

**General comments:**

1. I am not able to follow the modeling framework. A separate section for the model description is necessary. The model derivation in detail and a list of variables and units should be also provided in the Supplement. Is the adjustment in photosynthetic machinery due to the changes in temperature common for Scots pine? To my knowledge, the photosynthetic machinery in some species (e.g., Pinus edulis, Juniperus monosperma and Pinus taeda) even did not acclimate after long-term manipulation of precipitation and atmospheric CO2 concentration. The authors also pointed out that acclimation is omitted in the proposed model. However, how acclimation occurred at a longer time-scale is differentiated from the short-term changes in photosynthetic machinery needs further explanation.

Response: The model was presented in detail in Hari, P., et al. (2017) Annual cycle of Scots pine's photosynthesis, Atmos. Chem. Phys., 17, 15045-15053, https://doi.org/10.5194/acp-17-15045-2017), and therefore only the extension to the GPP annual dynamics is explained here. A Table of parameters, units and fitted values was added (Table 1).

It is evident that short-term acclimation and longer-term adaptation to environmental drivers need to be defined very carefully and have fundamentally very different consequences for plant physiology and thus also in any modeling exercise. Short-term changes in photosynthesis machinery are acclimations while longer timescale should be defined as adaptations. In terms of trees, the adaptation to high CO2 would need some reproduction cycles and changes on genomic level. Even the longest FACE experiments are still too short term to achieve this. On any scale in space and time, we should see adaptation of physiological processes as a "mean" and acclimation as some "noise" or "deviation" around that mean. Larger scales enable us to get higher accuracy information on the mean behavior, while smaller scales are better assessing the acclimation processes.

Long-term acclimation to precipitation or CO2 were not considered here, as we focused on the temperature and irradiation as short-term drivers of the annual dynamics state of photosynthetic machinery. However, in some ecosystems, periodic drought may have similar impact on the photosynthetic machinery as we observe here. Short-term acclimation to higher CO2 has an impact on the activity and quantity of the functional units involved, but the basic underlying processes are unlikely to change dramatically with increased CO2. Therefore, the model can be used for predicting the longer-term acclimation as well.

2. To predict GPP across the five Scots pine stands from leaf-level model, a scaling coefficient was used to bridge the two largely separated spatial scales. The scaling coefficient for current year was estimated by data from previous year. This suggests that the scaling coefficient is dynamic (i.e., yearly). What would be the information from this yearly scaling coefficient? When the dynamic of photosynthetic machinery is only estimated from one site and subsequently used for the other four sites, how would you interpret the differences in the scaling coefficients across the five sites?

Response: We assume that the scaling coefficient is not dynamic but rather stable within the site and characterizes the stand structure in an aggregated manner. We estimated the value from previous year to avoid estimation bias in the fit. We use the same parameter values for all sites. The differences in the parameter values is an additional source of variation in the value of the scaling factor.

3. I am not sure if the proposed model can accommodate the effects of water-stressed condition in the soil on stomatal conductance especially when the authors mainly focus on the responses to light and CO2. In fact, how the differences in the environmental factors impact the behavior of GPP across the five sites is not discussed in the manuscript. If water-stressed condition in the soil is not explicitly considered in the leaf-level model, do we expect that this information is embedded in the scaling coefficient?

Response: The setup of the model from theory incorporates the water stress in the optimal stomatal control. In that sense, using data to find parameters should yield in a set that has found the optimal stomatal control for a certain photosynthesis value in that case. Using a very large amount of data (large temporal scale even on leaf level) a "mean" optimal value should be found. This includes now also evidently local drought periods etc. This large temporal scale value is the one we use further on the ecosystem level.

**Specific comments:**

1. P2L22 Definition of stable regularities is needed.

Response: With 'stable regularities' we mean here the fundamental and simple processes which can be scaled from smaller units to larger regions over the whole continent. Our approach is to develop here a model, which is robust enough to be applied for the annual variations of photosynthesis with very small number of environmental parameters and widely different climatic conditions. ....

2. The order of Fig. 1 and 2 should be corrected to match the main text.

Response: Figures 1 and 2 have been changed in the text and their order of appearance as well.

3. Comparison between measured and modeled S (i.e., the state of photosynthetic machinery) as well as related discussions should be provided.

Response: The details in the model and especially the fundamental concept of the annual cycle has been published in Hari, et al. (2017) Annual cycle of Scots pine's photosynthesis, Atmos. Chem. Phys., 17, 15045-15053, https://doi.org/10.5194/acp-17-15045-2017),

4. P4L25 Description of up-scaling processes and the calculated scaling coefficient should be reported. Interpretation for the scaling coefficient is also required

Response: The scaling coefficient incorporates all differences between sites, such as differences in fine structure of leaves, differences in the photosynthetic machinery, (concentrations of pigments, membrane pumps and enzymes), differences in canopy structure often described with LAI, site fertility and others. See also the response to Ref 1, Q1.

5. P5L13-16 Is it possible that the photosynthetic parameters for the five sites are actually different but this information is embedded in scaling coefficient?

Response: We have described the scaling coefficient in our response to Ref 1, Q1. In brief, the GPP and also photosynthesis are scaling (and adapted) with temperature. The "mean" set of photosynthetic parameters will also scale with that. In northern ecosystems, less variability is observed in the "acclimation" to the specific stressors (drought, high light, etc.) and in southern ecosystems the deviations or "acclimations" are higher, generating more noise. That all can be said from the nature of the equation describing S.

6. P5L27-29 There are many models that can be used to predict stomatal conductance and then photosynthetic  $CO_2$  flux in response to different environmental factors.

Response: Our purpose in this paper is to show that in order to predict the annual dynamics in photosynthesis, both stomatal conductance and the physiological processes related to the inherent carbon assimilation and light adsorbance, and - essentially - their synchronized assimilation to the system are needed. Our model includes the optimal stomatal control as the main factor for determining the CO2 diffusion to the site of photosynthesis. We have added more explanation on this aspect on p. 3 (lines 10-14).

7. Discussion regarding different environmental conditions (e.g., temperature, precipitation, soil water status...) across the five sites should be included. To do so, time series of environmental factors for the five sites should be also provided when comparing the measured with predicted GPP (i.e., fig 1 or 2?).

Response: The analysis of residuals (Fig 4) gives a good view of the range of conditions at each site during the measurements. It also shows the comparison between the GPP (modeled and predicted with the model). Therefore we did not add any new figure for this purpose. See also our responses to Scaling issues (Ref 1, Q1 and Ref 2, Q5)

**Prediction of photosynthesis in Scots pine ecosystems across Europe by a needle-level theory**

Pertti Hari1, Steffen Noe2, Sigrid Dengel3, Jan Elbers4, Bert Gielen5, Tiia Grönholm6, Veli-Matti
 Kerminen6, Bart Kruijt4, Liisa Kulmala1, Samuli Launiainen7, Anders Lindroth8Lindroth7, Ivan Mammarella6, Tuukka Petäjä6, Guy Schurgers9Schurgers8, Anni Vanhatalo1, Timo Vesala4,6, Markku Kulmala6, and Jaana Bäck1

1University of Helsinki, Institute for Atmospheric and Earth System Research INAR, Faculty of Agriculture and Forestry, Department of Forest Sciences, P.O. Box 27, FI-00014 University of Helsinki, Finland

2Estonian University of Life Sciences, Institute of Agricultural and Environmental Sciences, Department of Plant Physiology, Kreutzwaldi 1, EE-51014 Tartu, Estonia
 3Lawrence Berkeley National Laboratory, Climate and Ecosystem Sciences Division, 1 Cyclotron Road 84-155, Mail Stop 074-0316, Berkeley, CA 94720-8118, USA
 4WWageningen University and Research, ageningen Environmental Research, P.O. Box 47, 6700AA Wageningen, Netherlands
 5University of Life Content of Division 2010 Wildiil, D.L.F.

- 5University of Antwerp, Department of Biology, 2610 Wilrijk, Belgium

[revised manuscript text omitted]

---

## Author Response (AR2)

**Author response for**

*Prediction of photosynthesis in Scots pine ecosystems across Europe by needle-level theory*

**by Pertti Hari et al. in Atmos. Chem. Phys. Discuss.**

Dear Dominick,

Thank you for the comments. The main comment of the reviewer 1, the scaling issue, is showing a difference at a conceptual level between a process-based model and EC measurement oriented research, and we have tried to clarify our view below in red.

Reviewer 1:

General comments:

The clarity and organization of the manuscript have been improved after the revision made by the authors. However, the authors did not take all the comments from both reviewers into account and explain their rationale and actions well. It is for this reason that I do not suggest publications in ACP unless the authors can clearly explain what is the information embedded in the ecosystem-specific scaling coefficient when upscaling the leaf-level model to ecosystem-scale GPP.

Response: To make it easy, any measure on a specific scale reflects an integrated or averaged value at that certain scale. Any change in scale needs a "translating" coefficient and the direction, up or down, does not matter. It seems usually more "natural" to upscale from a more detailed to a more aggregated scale than to downscale. However, changing a scale implies always a loss of information, either on details or on the aggregated state. Therefore, it seems a bit awkward that the reviewer wants a "clear explanation" on the "information embedded in the … scaling coefficient". If there would be such a clear explanation, it would be sufficient to just sum up in the case of upscaling, no need for any coefficient that represents the "loss of information". In the best case, we can assume that the scaling coefficient carries some information (=facts that we know about something) on the average noise gain (=loss of information) due to the scaling.

A note on the use of the term "upscaling": We used this term in our original manuscript only once, in the introduction and referring to studies done by other researchers. We stated "The upscaling from leaf to ecosystem scale is done either using 'big-leaf' approaches (dePury and Farquhar, 1997; Wang and Leuning, 1998), or by incorporating the impacts of 20 vertical canopy structure on microclimatic drivers, solar radiation in particular, via multi-layer models of different complexity (Leuning, 1995; Baldocchi and Meyers, 1998)". This does not imply that we used the exact approach as given in the literature cited.

In the latest manuscript, in the two paragraphs following the above citation we draw the attention of the reader to the GPP operating on the scale of the whole ecosystem. We acknowledge the processes in the underlying scales and finally point out: "This common functional basis generates common regularities in the behaviour of photosynthesis". Therefore, our main assumption is, that the regularities are emergent and on a common basis, i.e. scale free. They have to be apparent on all levels of organization that make up the complex adaptive system we look at.

In the final sentence of the introduction, we express our aim: "The aim of our paper is to study the role of these regularities in the behaviour of the photosynthetic $CO_2$ flux, observed in the measurements at one site, Värriö, and use the above concepts to analyse the EC flux data in several Scots pine stands across Europe (Fig. 1)".

These two statements tell, that we aim, in the ideal case, for a scale-free, abstract theory that explains photosynthetic behavior on ecosystem level (GPP). Comparison of the GPP across different sites of Scots pine "ecosystems" over Europe let us grade the prediction power of that theory. The novel finding in our theoretical

work is that the temperature (S parameter) explains on a longer temporal scale (annual) the largest portion of the seasonal change at a site and a minor effect measures the growth of noise (site specific scaling coefficient) according structural differences at different sites. However, the noise grows also consistently with the mean temperature at the different sites, lowest in northernmost Värriö, highest in southernmost Brasschaat.

1. The soil drying process can be considered by the leaf-level gas exchange model proposed elsewhere (Hari et al., 2017) through the model parameter λ (i.e., the cost of transpiration; a measure of water use efficiency). Based on the reply to previous review, however, only a "mean" optimal value (inferred from a large corpus of data) is used. This suggests that how leaf-level gas exchange is impacted by the soil water status is not explicitly taken into account.

Response: We acknowledge that the impact of dry soil conditions on GPP is large. However, this paper describes the leaf level model applied at ecosystem scale, and does not aim at specifically addressing the drought conditions. The use of EC data also includes ultimately all the factors the ecosystem is facing, including potential drought conditions. Therefore, any drought condition is "implicitly" taken into account. We do not see the value for an explicit expression without explicit data on the ecosystem scale. Soil water potential is in EC data implicit included, not explicit.

2. The up-scaling is mainly done through LAI of the dominant species (i.e., Scot pine). However, the ecosystem-level GPP can be also impacted by the activities of understory species and the soil. When adding an extra parameter (i.e., ecosystem-specific scaling coefficient) to upscale leaf-level processes to ecosystem level, this scaling coefficient then contains information associated with all the possible contributions other than leaf-level processes from single species. This explains why both reviewers suggested that the ecosystem-specific scaling coefficient should be reported and further discussed especially when the soil-drying effects are not included in the leaf-level model.

Response: We agree with the reviewer that the site-specific "scaling" coefficient is aggregating many structural and physical features. But, where does the reviewer come to the claim of the upscaling by LAI of the dominant species? We did not write that in either version of the manuscript. In the answers to the reviewers of the first version, we discussed also the link between temperature LAI and photosynthetic efficiency. Furthermore, the residual analysis showed a large independency from LAI.

Again, the use of EC data includes also all these site-specific features, including understory species, soil features, stand density etc. We have therefore assumed that we do not need to explicitly introduce them here. Please revisit the text written before on "information" and "scaling coefficients", there can be no other information in a scaling coefficient than a statistical measure of information loss, anything else is assumption. We discussed from the beginning that the "noise" on ecosystem level measured GPP originates from "omitting structural details".

Specific comments:
1. p2, Line 24-30: Duursma et al. (2009) also did not consider drying soil effects in their model.

Response: We do not say they did. By adding the Duursma et a.l reference we wish to show that they also suggest a similar seasonality as in our model.

2. Eqs. (1) and (3): Is E the transpiration rate or efficiency of photosynthetic light and carbon? According to Hari et al. (2017), it seems like that E is the same as b. What is difference between Eq (1) and (3)? Is b=a4S=E or there is time up-scaling from daily p (i.e., Eq (1)) to annual p (i.e., Eq. (3))?

Response: The difference between Eq (1) and Eq (3) is that the first one describes the daily and the second one the annual GPP. E is transpiration rate and b is the efficiency of photosynthesis. On annual scale $b=a_4*S$.

3. p5, Line 14-20: This is why the ecosystem-specific scaling coefficient should be reported and carefully discussed.

Response: Our theoretical approach in this paper is based on the Hari et al (2017) model, developed and parameterized at leaf level, and, based on our analysis, it seems to be rather well scalable to a longer time scale (annual) as well as from leaf to stand scales. This is a novel and surprising result and to us it clearly shows the power of a seemingly very simple model. We do not here extend the analysis into a more detailed analysis of the scaling coefficient, but we recognize that it is possible to do that with more detailed site-specific data on conditions and structure of the stand, if available.

However, it should be once more noted that we aimed to show differences between Scots pine sites over Europe on ecosystem scale and an annual temporal scale, not to fit some local parameterization of photosynthesis or GPP according drought.

4. Fig. 3 A): Should it be the onset dates of photosynthesis not the onset and cessation dates of …..?
Response: Yes, the reviewer is correct. This was a typo remaining from a previous version of the manuscript, thanks for noticing!

5. p6, Line 22-29: Is the residual calculated as the difference between the measured and calculated GPP? If so, the same order of residual as measured GPP suggests significant deviation between predicted and measured GPP. This may not be surprised especially when the variation of GPP is highly non-linear with respect to the environmental factors (e.g., T, CO2, PAR and soil water status). That is, the residual for a specific environmental category can be still impacted by other environmental factors.

Response: The residuals are calculated as the difference between measured and modeled GPP. Figure 4 shows that there is very little any systematic behavior when plotted as a function of environmental factors. It is clear that some of them (e.g., air temperature and PAR) are co-varying, especially over longer time scales, and thus the impact of a single factor is difficult to separate.